# Preanalytical Pitfalls in Untargeted Plasma Nuclear Magnetic Resonance Metabolomics of Endocrine Hypertension

**DOI:** 10.3390/metabo12080679

**Published:** 2022-07-24

**Authors:** Nikolaos G. Bliziotis, Leo A. J. Kluijtmans, Gerjen H. Tinnevelt, Parminder Reel, Smarti Reel, Katharina Langton, Mercedes Robledo, Christina Pamporaki, Alessio Pecori, Josie Van Kralingen, Martina Tetti, Udo F. H. Engelke, Zoran Erlic, Jasper Engel, Timo Deutschbein, Svenja Nölting, Aleksander Prejbisz, Susan Richter, Jerzy Adamski, Andrzej Januszewicz, Filippo Ceccato, Carla Scaroni, Michael C. Dennedy, Tracy A. Williams, Livia Lenzini, Anne-Paule Gimenez-Roqueplo, Eleanor Davies, Martin Fassnacht, Hanna Remde, Graeme Eisenhofer, Felix Beuschlein, Matthias Kroiss, Emily Jefferson, Maria-Christina Zennaro, Ron A. Wevers, Jeroen J. Jansen, Jaap Deinum, Henri J. L. M. Timmers

**Affiliations:** 1Department of Laboratory Medicine, Translational Metabolic Laboratory, Radboud University Medical Center, 6525 GA Nijmegen, The Netherlands; udo.engelke@radboudumc.nl; 2Department of Analytical Chemistry, Institute for Molecules and Materials, Radboud University, 6500 HB Nijmegen, The Netherlands; g.tinnevelt@science.ru.nl (G.H.T.); jeroen.jansen@ru.nl (J.J.J.); 3Division of Population Health and Genomics, School of Medicine, University of Dundee, Dundee DD2 4BF, UK; p.s.reel@dundee.ac.uk (P.R.); s.reel@dundee.ac.uk (S.R.); e.r.jefferson@dundee.ac.uk (E.J.); 4Department of Medicine III, University Hospital Carl Gustav Carus, Technische Universität Dresden, 01307 Dresden, Germany; katharina.langton@uniklinikum-dresden.de (K.L.); christina.pamporaki@uniklinikum-dresden.de (C.P.); graeme.eisenhofer@uniklinikum-dresden.de (G.E.); 5Hereditary Endocrine Cancer Group, Spanish National Cancer Research Centre (CNIO), Centro de Investigación Biomédica en Red de Enfermedades Raras (CIBERER), 28029 Madrid, Spain; mrobledo@cnio.es; 6Division of Internal Medicine and Hypertension Unit, Department of Medical Sciences, University of Torino, 10124 Torino, Italy; alessio.pecori@edu.unito.it (A.P.); tetti.martina@gmail.com (M.T.); tracyannwilliams48@gmail.com (T.A.W.); 7British Heart Foundation Glasgow Cardiovascular Research Centre (BHF GCRC), Institute of Cardiovascular & Medical Sciences (ICAMS), University of Glasgow, Glasgow G12 8TA, UK; josievankralingen@googlemail.com (J.V.K.); eleanor.davies@glasgow.ac.uk (E.D.); 8Department of Endocrinology, Diabetology and Clinical Nutrition, University Hospital Zurich (USZ), University of Zurich (UZH), 8006 Zurich, Switzerland; zoran.erlic@usz.ch (Z.E.); felix.beuschlein@usz.ch (F.B.); 9Biometris, Wageningen University & Research, 6708 PB Wageningen, The Netherlands; jasper.engel@wur.nl; 10Department of Internal Medicine I, Division of Endocrinology and Diabetes, University Hospital, University of Würzburg, 97080 Würzburg, Germany; deutschbein_t@ukw.de (T.D.); fassnacht_m@ukw.de (M.F.); remde_h@ukw.de (H.R.); kroiss_m@ukw.de (M.K.); 11Medicover Oldenburg MVZ, 26122 Oldenburg, Germany; 12Department of Medicine IV, University Hospital, LMU Munich, 80336 Munich, Germany; svenja.noelting@med.uni-muenchen.de; 13Department of Hypertension, Institute of Cardiology, 04-628 Warsaw, Poland; prejbisz@vp.pl (A.P.); ajanu@op.pl (A.J.); 14Institute of Clinical Chemistry and Laboratory Medicine, University Hospital Carl Gustav Carus at the Technische Universität Dresden, 01307 Dresden, Germany; susan.richter@uniklinikum-dresden.de; 15Research Unit Molecular Endocrinology and Metabolism, Genome Analysis Center, Helmholtz Center München, German Research Center for Environmental Health, 85764 Neuherberg, Germany; adamski@helmholtz-muenchen.de; 16Institute of Biochemistry, Faculty of Medicine, University of Ljubljana, 1000 Ljubljana, Slovenia; 17Institute of Experimental Genetics, Technical University München, 85350 Freising-Weihenstephan, Germany; 18Department of Biochemistry, Yong Loo Lin School of Medicine, National University of Singapore, 119077 Singapore, Singapore; 19Endocrinology Unit, Department of Medicine DIMED, University-Hospital of Padova, 35128 Padova, Italy; ceccato.filippo@gmail.com (F.C.); carla.scaroni@unipd.it (C.S.); 20The Discipline of Pharmacology and Therapeutics, School of Medicine, National University of Ireland, H91 CF50 Galway, Ireland; conall.dennedy@mailn.hse.ie; 21Department of Medicine-DIMED, Emergency and Hypertension Unit, University of Padova, University Hospital, 35126 Padova, Italy; livia.lenzini@gmail.com; 22INSERM, PARCC, Université de Paris, 75015 Paris, France; anne-paule.gimenez-roqueplo@aphp.fr (A.-P.G.-R.); maria-christina.zennaro@inserm.fr (M.-C.Z.); 23Service de Genétique, Assistance Publique-Hôpitaux de Paris, Hôpital Européen Georges Pompidou, 75015 Paris, France; 24Core Unit Clinical Mass Spectrometry, University Hospital Würzburg, 97080 Würzburg, Germany; 25Comprehensive Cancer Center Mainfranken, Würzburg University, 97070 Würzburg, Germany; 26Institute of Health & Wellbeing, Glasgow University, Glasgow G12 8RZ, UK; 27Department of Internal Medicine, Radboud University Medical Center, 6525 GA Nijmegen, The Netherlands; jaap.deinum@radboudumc.nl

**Keywords:** confounders, metabolomics, multicenter, plasma NMR, preanalytical conditions

## Abstract

Despite considerable morbidity and mortality, numerous cases of endocrine hypertension (EHT) forms, including primary aldosteronism (PA), pheochromocytoma and functional paraganglioma (PPGL), and Cushing’s syndrome (CS), remain undetected. We aimed to establish signatures for the different forms of EHT, investigate potentially confounding effects and establish unbiased disease biomarkers. Plasma samples were obtained from 13 biobanks across seven countries and analyzed using untargeted NMR metabolomics. We compared unstratified samples of 106 PHT patients to 231 EHT patients, including 104 PA, 94 PPGL and 33 CS patients. Spectra were subjected to a multivariate statistical comparison of PHT to EHT forms and the associated signatures were obtained. Three approaches were applied to investigate and correct confounding effects. Though we found signatures that could separate PHT from EHT forms, there were also key similarities with the signatures of sample center of origin and sample age. The study design restricted the applicability of the corrections employed. With the samples that were available, no biomarkers for PHT vs. EHT could be identified. The complexity of the confounding effects, evidenced by their robustness to correction approaches, highlighted the need for a consensus on how to deal with variabilities probably attributed to preanalytical factors in retrospective, multicenter metabolomics studies.

## 1. Introduction

Arterial hypertension (HT) was prevalent in approximately 22% of the global population in 2015 [1] and was found to be the leading cause of death in 2017 [2]. Three types of secondary HT, specifically endocrine HT (EHT), are primary aldosteronism (PA), pheochromocytoma and paraganglioma (PPGL), and Cushing’s syndrome (CS). All these diseases are associated with increased morbidity and mortality [3,4,5,6,7]. Tailored medical treatments, in specific surgical resection of the underlying hormone-producing lesion, result in decreased morbidity and mortality of patients [3,5,6,7], therefore correct diagnosis is key for patient survival. However, diagnosing EHT can be challenging, due to the dependence of the PA screening test on numerous factors [8], highly variable clinical presentations and a lack of routine cost-effective biochemical screening for PPGLs [9], and heavy reliance on medical experts’ experience in CS screening [10,11]. As a result, many cases of PA, PPGL and CS remain undetected [8,11,12]. By employing a more general, objective and cost-effective screening method, the chances for a long-term cure as well as blood pressure control should be increased [13]. In addition, early disease identification may shed light on pathophysiological mechanisms allowing for personalized treatment approaches [13]. Metabolomics has led to the discovery of potential biomarkers for several diseases [14,15,16,17]. The ENSAT-HT consortium aims to establish multi-omics diagnostic biomarkers, including metabolomics, for classifying patients as having either primary HT (PHT) or a form of EHT [18]. Erlic et al. [19] applied a targeted Liquid Chromatography–Mass Spectrometry approach to differentiate ENSAT-HT plasma samples collected from patients with EHT from those collected from patients with PHT and delineated a biomarker signature with high classification accuracy.

Metabolomics studies, however, when applied to clinical studies, are prone to limitations such as confounding effects [20]. For example, the initial report on the high accuracy of metabolomics in predicting coronary artery disease [21], was contrasted by the follow-up report by Kirschenlohr et al., in which patient gender and medication were identified as sources of unwanted variation and lower accuracies were found by analyzing subgroups [22]. According to the 2005 review by David Ransohoff, “Bias can be so powerful in non-experimental observational research that a study should be presumed ‘guilty’—or biased—until proven innocent” [23], therefore it is advisable to investigate and correct the influence of confounders in metabolomics studies. Confounding effects can be corrected using methods such as Analysis Of Variance Simultaneous Component Analysis (ASCA) [24], Covariate-Adjusted Projection to Latent Structures [25] and regularized Multivariate Analysis Of Variance [26].

In this paper, we analyzed bio-banked plasma samples sent from various centers that are part of the ENSAT-HT consortium, using proton nuclear magnetic resonance spectroscopy (^1^H-NMR)-based untargeted metabolomics as an alternative and possibly complementary approach to targeted metabolomics [19]. We aimed to establish unbiased EHT biomarkers, so we initially compared each EHT disease group to PHT and obtained signatures of each comparison from the analyses. Subsequently, we applied three approaches to investigate and correct the influence of confounders on the obtained signatures, as center heterogeneity is a well-known challenge in multicenter studies [27], particularly when disease groups are inadequately represented in each center’s patient cohort [28].

## 2. Results

### 2.1. Initial Approach: Establishing Possible EHT–PHT Biomarkers

Table 1 summarizes the characteristics of all patient samples analyzed. Briefly, our final dataset consisted of 231 plasma samples collected from patients with EHT (104 PA + 94 PPGL + 33 CS) and 106 samples from patients with PHT.

Results from the peak picking procedure are shown in Figure 1.

The Principal Component Analysis (PCA) score plot of the complete dataset including all 86 peaks (Appendix A) showed Quality Control (QC) samples clustering closely together, indicating a low analytical variation compared to the sum of biological and preanalytical variation. The result shown in Figure 2a appeared promising for the successful separation of the two main disease groups (EHT and PHT), as the two disease groups resulted in slightly different scores along the second principal component (PC2). The classification analyses of these datasets resulted in high accuracies of approximately 80% when attempting to distinguish PHT from EHT forms (Appendix A and Table 2). Lower balanced accuracies (e.g., 65%, Table 2) resulted from separating all groups from each other. Table 3 depicts the signatures obtained from scenario EHT–PHT.

By investigating the various sources of variation present in our data using PCA, sample center of origin, as well as sample age, appeared to be major sources of variation in the data. As illustrated in Figure 2b, clusters of centers were apparent. Figure 2c demonstrates a tendency for separation of samples based on their sample age, which was significantly different between disease groups, as well as amongst centers, according to a Kruskal–Wallis test (*p* < 2 × 10^−16^). Effects of other factors, such as analytical batch and patient age, were not observed to coincide with tendencies in the first two principal components (Appendix A). Inspection of the corresponding loadings plot (Figure 2d and Appendix A) showed that, in comparison to the cluster of centers Dresden (GYDR), Lübeck (GYLU), München (GYMU), Würzburg (GYWU), Torino (ITTU3), Nijmegen (NLNI) and Warsaw (PLWW), samples in cluster Paris (FRPA1) PHT had low glutamine and high glutamate, whereas samples from centers FRPA1 (non-PHT), FRPA2, Glasgow (GBGL2), Galway (IRGA) and Padova (ITPD/ITPD3) were characterized by high lactate and ornithine, GBGL2 samples were further characterized by high glutamate and low glutamine, and ITPD/ITPD3 samples had high methanol. These metabolites, as well as those found in signatures, were identified as described in the Methods section of this paper and shown in Figure 3.

### 2.2. Correcting for Confounders: Approach A (ASCA Correction)

We could not use Analysis of Variance Simultaneous Component Analysis (ASCA) to correct for the center directly, as there was not an adequate representation of each center in all investigated groups. Therefore, we attempted to correct our data for possible confounders using ASCA, performed according to the observed clustering of centers along the first principal component (Figure 2b, with centers GYDR, GYLU, GYMU, GYWU, ITTU3, NLNI, PLWW, as well as the FRPA1 PHT group included in cluster 1 and FRPA1 (non-PHT), FRPA2, GBGL2, IRGA, ITPD/ITPD3 in cluster 2. We did not use ASCA to correct for sample age, as our implementation did not allow continuous variables to be considered as design factors. As shown in Appendix A and Table 2, classification analyses after using ASCA for correcting the data resulted in overall similar results to the previous approach. ASCA correction was not possible on the CS-PHT or the all vs. all dataset, as there were no samples represented by cluster 1 in the CS group.

According to Table 3, glycine and lactate were the differences between the ASCA and Initial Approach signatures of scenario EHT-PHT, indicating that their levels were normalized by ASCA, diminishing their importance. In the other scenarios, ASCA similarly normalized levels of metabolites that correlated to the first principal component of Figure 2.

### 2.3. Correcting for Confounders: Approach B (Metabolite Exclusions)

As shown in Figure 2, center and sample age appeared to be major sources of variation in our data. Appendix A provides an overview of all analyses carried out to assess the effects of these confounders on the various datasets. Overall, accuracies were higher when comparing groups defined by confounders than they were based on disease groups (Initial Approach). We employed PLSDA to establish the signatures of these effects (Table 4) and compared cluster 1 (excluding the FRPA1 PHT group due to a distinct signature) to cluster 2, as well as samples with a sample age below the median to those with a sample age above the median (calculated including all samples). We used PCA, due to limited sample sizes, to compare the FRPA1 PHT group to the rest of cluster 1. Metabolites acetylcarnitine, creatine, dimethyl sulfone, glucose, glutamate, glutamine, glycerol, glycine, lactate, methanol, methionine, ornithine, pyruvate and the unknown signal at 3.284 ppm were all found to be related to these confounders. In relation to disease groups, all scenarios resulted in signatures that included confounder-related metabolites. Notably, 9/13 metabolites making up the EHT–PHT signature depicted in Table 3 were also found to be related to a confounder.

Analyses of data after we excluded confounder-related peaks yielded lower accuracy estimates than those calculated from the complete datasets (Appendix A and Table 2).

Regarding signatures, Approach B resulted in a unique set of metabolites compared to the other approaches (with 8/12 metabolites not found with any other approach in the EHT–PHT scenario, Table 3). Some signals arising from the same metabolite were high in one group while others were high in the other, which may be the result of the low statistical power of the data, as peaks from the same metabolite correlated in the overall loadings plot in Figure 2d and Appendix A (except for the lysine peaks at 2.997 and 3.013 ppm). As a result, the resulting signatures were unreliable, though probably less affected by the confounding effects.

### 2.4. Correcting for Confounders: Approach C (Whole Center Exclusions)

As a final approach to illustrate the putative effects of confounders on our initial signature, we investigated disease group metabolome differences in the cluster which comprised only the centers shown in Table 5 (GYDR, GYLU, GYMU, GYWU, ITTU3, NLNI, PLWW), as these clustered closely compared to other centers (Figure 2b). We included a total of 40 patients with PHT, along with 118 patients with EHT. Of the latter, 54 had PA and 64 had PPGL.

Though center selection led to lesser confounding effects compared to the Initial Approach, it also led to lower accuracies for all investigated scenarios. Appendix A and Table 2 summarize the resulting accuracies obtained from this approach; accuracies of approximately 60% were found when comparing EHT to PHT, or PA–PPGL–PHT, and approximately 70% via the PA–PHT and PPGL–PHT scenarios. When predicting the excluded samples’ disease groups using these models, the EHT–PHT model was 56% accurate, with a sensitivity of 26% and a specificity of 86%, the PA–PHT model was 50% accurate, with a sensitivity of 6% and a specificity of 94%, and the PPGL–PHT model was 65% accurate, with a sensitivity of 57% and a specificity of 74%. To evaluate any remaining confounding effects, we also compared center GYDR to ITTU3, resulting in higher accuracies than the analyses comparing disease groups (Appendix A), indicating the possible presence of an additional source of center-related bias. Considering whole center removal as the most effective way of diminishing the influence of the possible confounders seen in Figure 2, we used the Approach C signature to evaluate the other three. According to Table 3, glutamine and glutamate were not found to be related to disease group separation by Approach C, contrasting both the Initial Approach and Approach A. Additionally, seven confounder-related metabolites, which were excluded for Approach B, were selected by Approach C as predictors. Another group of seven metabolites selected only by Approach B was not corroborated by Approach C. Similar observations made in the other scenarios comparing approaches show that Approach A did not correct for some effects at all, several metabolites were related to both confounders and disease groups, and Approach B may have led to overfitted results.

## 3. Discussion

We found that our untargeted NMR metabolomics signatures of endocrine hypertension were statistically related to confounders. The metabolomes of our samples were organized in distinct clusters, primarily defined by their center of origin. Glutamate, glutamine, lactate, ornithine and methanol appeared to have a strong relation to this center-related clustering, while also being a part of the EHT–PHT signature. In the targeted metabolomics study [19], both glutamate and glutamine were included in the reported biomarkers. Glutamine was reported to correlate positively to atherogenesis [29,30], and high levels of the amino acid were hypothesized to protect vasculature [31]. Lower levels of the amino acid were found in hypertensive men compared to healthy controls [15], whereas higher levels were found in hypertensive women compared to controls [32]. Glutamate was included in a panel for detecting albuminuria in hypertensive patients [33] and, along with proline, has shown a statistically different concentration distribution in hypertensive nephrosclerosis compared to controls [34]. In our previous work [35], proline and methanol were found to be higher in preoperative PPGL patients. To investigate possible confounders in our EHT–PHT signatures, we employed three correction approaches to compare results. We used ASCA [24] (Approach A) but found full correction of possibly confounding effects impossible, due to the lack of adequate representation of each center in each disease group.

We, therefore, established a signature for the center-related clustering, to formulate a hypothesis as to the reason for this source of variation and to determine its potentially confounding effect on the initially established EHT–PHT signature. We found lactate increased and glucose decreased in samples originating from cluster 2 centers compared to those from cluster 1, which was found before in plasma samples harvested from whole blood after a pre-centrifugation delay due to prolonged red blood cell glycolysis [36,37,38,39,40,41,42,43,44,45,46,47]; this would mean that cluster 2 included samples collected after such a delay, compared to the rest of the samples. The coinciding lower levels of pyruvate in cluster 2 compared to cluster 1 samples were shown before when there is a delay in cold temperatures [37,38,42,44,46] (Brunius et al. report the contrary [39]), whereas, at room temperature, pyruvate was found to increase [37,39,40,44,46] (contrary results in [42]). A delay before whole blood centrifugation could also explain the lower levels of glutamine in cluster 2 compared to cluster 1 [40,43] and higher glutamate [40,46,47,48] (contrary results in [42]), possibly due to the conversion of glutamine to glutamate by glutaminase [48], and higher ornithine [40,42,46,47], possibly due to the activity of erythrocyte arginase [40]. Interestingly, cluster 2′s higher glycerol is not supported by the literature [40,42], in which it is reported to be lower in samples collected after a pre-centrifugation delay. It was unclear why methanol was higher in ITPD/ITPD3 samples compared to the rest, but it can be speculated that it could have originated as an impurity. The separate cluster of FRPA1 PHT samples can be explained by a delay between plasma harvesting and storage at room temperature, which would result in the observed increase in glutamate and decrease in glutamine, and in the lack of effects on glycolysis-related metabolite levels [42].

Sample age may have also influenced the resulting metabolic signature, especially given that there were significantly different sample ages amongst centers. We report results obtained from control (PHT) samples collected one year to 17 years before analysis (averaging at 6.5 years), and EHT samples ranging from less than two weeks to 9.5 years (averaging at 3 years), which show a similar sample age signature as that obtained from analyzing the metabolite differences between the two main clusters of samples. Though Pinto et al. [49] reported minimal metabolic impact of storage at −80 °C for a period up to 2.5 years, a recent study [50] reported increasing levels of glutamate and decreasing glutamine and pyruvate with storage time up to 16 years at −80 °C, as well as higher levels of ornithine, lactate and glycerol after 11 years of storage, in line with our results for these metabolites. Another study [51] found altered levels of several of these metabolites due to sample age. Specifically, increased levels of ornithine, methionine, glycine, glutamine and lower glutamate were found in samples analyzed after 5 years in −80 °C storage, of which we only found a similar direction for ornithine herein.

These findings, linking metabolites from the signatures obtained via the Initial Approach to any confounder, render our EHT–PHT signatures unreliable. In the targeted metabolomics study [19], which also used samples from the ENSAT-HT retrospective cohort, preanalytical confounders were not addressed, but GBGL2 samples were excluded.

The exclusion of confounder-related peaks (Approach B) resulted in analyses with lower statistical power, resulting also in low classification accuracies and signatures that could not be corroborated by any other approach. Metabolites such as lactate were excluded but were still found to play a role in discriminating samples by the final approach. Exclusion of whole centers (Approach C), specifically those we deemed compromised most by confounders, resulted in signatures that we used to evaluate the initially established ones. We expected this approach to yield the most reliable signature from our dataset and research questions, as it was the only one that both provided a complete view of the NMR plasma metabolome and kept confounding effects to a minimum by excluding compromised samples. Even so, we do not claim that Approach C signatures have merit in clinically discriminating PHT from any EHT form, as these analyses were restricted to certain centers and so included far fewer samples than originally planned. Approach C models predicted excluded PHT sample groups accurately, but sensitivity was low, while the highest accuracy achieved was 65% (PPGL-PHT model). Moreover, analyses comparing centers GYDR to ITTU3 were more accurate than those comparing disease groups in Approach C, which demonstrates another potentially confounding effect, albeit weaker than those already discussed. Approach C signatures differed substantially from the Initial Approach signatures, with several metabolites related to confounders not being selected as predictors or being selected with a coefficient of an opposite direction.

These differences underscore the uncertainty associated with the signatures from the Initial Approach, given the simultaneous relationship some metabolites have with both the confounders and the disease group separation. Furthermore, Approaches A and B do not seem to resolve the issue of confounding effects. Other approaches, such as multilevel analysis [52], which could be used to center the centers, would not be appropriate either, as EHT–PHT signatures varied by the center. Notably, the PA–PHT and PPGL–PHT signatures obtained from analyzing ITTU3 and GYDR data, respectively, were different from FRPA1 signatures (result not shown), as expected from the distinct cluster of FRPA1 PHT samples.

Untargeted NMR metabolomics of the ENSAT-HT retrospectively collected plasma samples described in this work was not suitable for obtaining a biomarker that discriminates EHT forms from PHT, due to the method’s sensitivity to the sample set’s probable confounders. Still, there were additional limitations in our work. Specifically, sample hemolysis [53] was not considered, but it was shown to not have a significant impact on the plasma metabolome with NMR [38], though the opposite was shown with MS [42]. Additionally, diet and medication at the time of sampling, as well as additional clinical parameters, such as patient BMI and disease severity, can all affect the metabolome [53] but were not considered here. There may also be additional confounders, e.g., analytical bias or patient characteristics, but their effect, if any, was less than that of the confounding effects already addressed, since the latter was the strongest source of variation within our data. Methodologically, our NMR method was limited by a relatively high detection limit, resulting in 33 identified metabolites, of which we excluded the ketone bodies acetoacetate and 3-hydroxybutyrate due to their dependence on fasting levels as well as acetone, acetate and choline due to their dependence on run order. There were five peaks to which we could not assign any known metabolites, and four metabolites (dimethylamine, dimethylglycine, dimethyl sulfone and formate) were identified only by visual inspection and comparison to databases. Our internal standard, which was used for peak scaling, was recently found to be attenuated by the presence of macromolecules [54], possibly resulting in lower peak intensities in samples with high concentrations of macromolecules. However, given that in our comparison of methods in our own paper 19/25 (76%) peaks in the final (LED) signature were also found in the ultrafiltration signature [55], we concluded that any bias, e.g., from the binding of maleic acid to protein, represented a minority of variation in a dataset, compared to the differences between groups. Permutation testing, though invaluable for assessing model accuracy, was not performed for the models presented, as these were deemed confounded and their resulting biomarkers unreliable.

## 4. Materials and Methods

### 4.1. Patient and Sample Characteristics

We analyzed Lithium Heparin blood plasma samples collected from 356 patients sampled at 13 ENSAT-HT centers, along with the post-operative counterparts of a subset of PPGL samples, which were used in our previous study [35]. After excluding technical outliers, 337 patients were finally included.

### 4.2. Untargeted 1H-NMR Metabolomics

We recorded spectra according to our NMR method as reported and previously applied [35,55]. Study samples, along with 146 Quality Control (QC) samples, were analyzed over 46 batches, whereas 99 Healthy Volunteer (HV) samples, were analyzed over 43 batches. A maximum of 15 samples per batch to limit intra-batch variability was selected, based on robust Principal Component Analysis (PCA) [56] results (R package “rospca” [57], version 1.0.4, Tom Reynkens), which showed QC samples outlying after 18 NMR experiments. NMR spectra recorded after this limit of 18 samples (*n* = 4, due to recording delays) were excluded. Samples were thawed and centrifuged at room temperature for the first three batches and at 4 °C for all the rest. A Bruker DRX AVANCE spectrometer equipped with a triple resonance inverse 5 mm probe head operating at 500.13 MHz was employed for analyzing samples. Both the sequence and batch number of all samples were recorded for estimating technical variability. Longitudinal Eddy-Current Delay (LED) spectra were recorded and processed as previously described [35,55]. Spectra with high line width were not used for further analysis (maleic acid peak width > 1.2 Hz, *n* = 4), along with spectra recorded from samples that had large chemical shift differences from other samples (*n* = 4) or peaks corresponding to EDTA (*n* = 2) or citrate (intensities corresponding to blood collection, *n* = 1). Four samples were retrospectively found to not correspond to any of the four disease groups (PA, PPGL, CS or PHT) and were excluded from the analysis. Areas corresponding to macromolecules, water, glucose and noise were excluded from data processing (areas above 10, 5.35 to 5.24, 5.15 to 4.40, 3.91 to 3.68, 3.54 to 3.36, 3.26 to 3.19, 1.30 to 1.10, 0.90 to 0.75, and below 0 ppm). R studio version 1.1.463 (J. J. Allaire, Boston, MA, USA) [58] running R version 3.4.4 (R core team, Vienna, Austria) [59] was used for loading the R package “batman” [60] version 1.2.1.03 (Jie Hao), which was used to extract the spectra as tables. R package “SPEAQ” [61] version 2.0.0 (Charlie Beirnaert) was used for obtaining the peak table and was applied according to the script “SPEAQ pipeline 3”. Due to their high number, SPEAQ was not possible to perform on all samples simultaneously. Hence, a script for aligning different SPEAQ batches was developed (“Combining1 2 PLASMA 3 new”). These two scripts can be found on the first author’s GitHub page (https://github.com/NickBliz/PPGL-PRE-VS-POST, last update on 5 June 2021) and data can be provided upon request. All subsequent data processing steps, which can be found in scripts on the first author’s GitHub page (https://github.com/NickBliz/ENSAT-HT-PLASMA-NMR, last update on 28 September 2021), were performed on R version 3.6.3, loaded on R studio version 1.2.5033. The ethanol peaks at 3.65 ppm, the highly variable histidine peaks at 7.06 and 7.81 ppm, the macromolecule peak at 2.02 ppm and the peaks with a strong correlation with the run order of samples at 3.185, 2.21 and 1.904 ppm were also excluded, along with peaks at 2.26, 2.28, 2.31, 2.37, 2.39 and 4.13 ppm, corresponding to ketone bodies. These latter metabolites were not taken into account as, giving rise to outliers, would not be valuable biomarkers. HV samples could be separated based on the batch in which they were analyzed, and QC samples based on their run order within batches, but these analytical factors were not found to have a major impact on our dataset, as seen in Appendix A. As the first step in data processing, features not present in at least 80% of samples belonging to either the QC or HV group [62], were removed. Probabilistic Quotient Normalization (PQN [63]) was applied as a normalization method, using as reference the median spectrum ignoring non-detects of a set of samples that were drawn from 98 HVs from the GYDR center. HV samples obtained from center FRPA1 were not used for normalization, as these only included healthy young male subjects. Next, peaks with a coefficient of variation of more than 30% in QC samples were removed. Finally, after missing value estimation with the k-nearest neighbors (k-NN) method [64] (k = 10), using R package “impute” version 1.58.0 (Balasubramanian Narasimhan) [65], the generalized log transformation (GLOG [66]) based on 133 QC samples was applied using R package “LMGene” version 2.40.0 (Blythe Durbin-Johnson) [67]. The QC samples were prepared by pooling plasma obtained from 390 anonymized plasma samples to a total volume of about 450 mL, which was subsequently aliquoted into 1 mL batches and stored at −80 °C until analysis. As a normalization and GLOG transformation optimization step, the process was repeated after the HV and QC sample groups were relieved of outliers detected by means of robust PCA [56]. The resulting dataset was directly used for multivariate statistics. We assigned peaks to metabolites using Chenomx evaluation v. 8.4 [68] and Bruker Topspin v. 4.0.6. We used the human metabolome database [69], along with the Madison Metabolomics Consortium Database [70], as references. To aid in peak assignment, additional methods were employed. Specifically, 2D NMR, namely correlation spectroscopy (COSY, cosyprqf) and J-resolved (JRES, jresgpprqf) pulse sequences allowed for investigating correlations between peaks and peak multiplicity, respectively. These experiments, along with 1D NMR on filtered samples in pH 2.5 [71], allowed for a number of peak identities to be confirmed, and some unknowns to be assigned to metabolites. COSY spectra were recorded using 4 K data points in F2 direction, 128 in F1, 16 scans, 8 dummy scans, a spectral width of 6 K Hz in both directions, an acquisition time of 0.02 s in F2 and 0.34 s in F1, resolution of 46.89 Hz in F2 and 2.83 Hz in F1. Receiver gain was set to the optimum value. Spectra were processed using Fourier transform in both directions and calibrated Trimethylsilylpropanoic acid (TSP) to 0 ppm. JRES spectra were recorded using 8 K data points in F2 direction, 64 in F1, 16 scans, 8 dummy scans, a spectral width of 8 K Hz in F2 and 60 in F1, an acquisition time of 0.49 s in F2 and 1.07 s in F1, resolution of 2.03 Hz in F2 and 0.94 Hz in F1. Receiver gain was set to the optimum value. Spectra were processed using Fourier transform in both directions, tilting and symmetrizing phase-sensitive spectra, and calibrated TSP to 0 ppm. F Furthermore, a spiking experiment was carried out, utilizing a QC sample and a non-related-to-the-study control sample, which were spiked with pyruvic acid, succinic acid, choline, carnitine, acetylcarnitine, methanol and 5-methylthioadenosine, along with histidine, serine, phenylalanine (Sigma) and tyrosine (Merck). A stock solution (#1) with concentrations of 0.31 mΜ of pyruvic acid, 0.15 mΜ succinic acid, 0.12 mΜ choline, 0.18 mΜ carnitine, 0.041 mΜ acetylcarnitine, 0.024 mΜ 5-methylthioadenosine and 0.99 mΜ methanol, another (#2) with 1.1 mΜ histidine, 1.5 mΜ serine, 0.60 mΜ phenylalanine, 0.42 mΜ tyrosine, along with two stock solutions (#3, #4) with half these concentrations, were prepared. A volume of 100 μL of stock #1 was added to 400 μL of a QC, another volume of 100 μL of stock #3 to another 400 μL of a QC, and a volume of 100 μL of stock #2 was added to 400 μL of the unrelated control, another volume of 100 μL of stock #4 to another 400 μL of the unrelated control. All these spiked samples underwent the same procedure as the study plasma samples to acquire as similar results as possible. Appendix A summarizes all peaks included for our analyses, along with their assigned metabolite names and the level of assignment rigor.

### 4.3. Data Analysis and Statistics

Based on our research aims, we aimed to separate the following disease groups: (a) EHT (PA + CS + PPGL) vs. PHT, (b) PA vs. PHT (c) CS vs. PHT, (d) PPGL vs. PHT, (e) PA vs. CS vs. PPGL vs. PHT. Multivariate Analysis (MVA) was performed on mean-centered data by employing the “MixOmics” R package version 6.11.33 (Kim-Anh Le Cao) [72] on R version 3.6.3, loaded on R studio version 1.2.5033. We used PCA [73], as an unsupervised method to discover trends and outliers within the data. Partial Least Squares Discriminant Analysis (PLSDA) [74] was used as a supervised method for establishing the signature for separating samples grouped by center as well as sample age (median sample age cutoff). Important PLSDA variables were determined based on their Variable Importance in the Projection (VIP) score [75] (median outer-loop VIP score above 1 after double cross-validation, or CV2) in PLSDA models. Analysis of Variance Simultaneous Component Analysis (ASCA [24]) was used for removing the interaction between PCA cluster and disease group, present within the dataset (as observed in the PCA score plot of Figure 2). Sparse PLSDA (sPLSDA [76,77]), was employed for investigating the metabolic signature (regression coefficients) of the differences between samples grouped by diagnostic category (PHT, PA, PPGL or CS, with EHT as a pool of the latter three and the first as controls). As an alternative to sPLSDA, l1-norm regularized logistic regression [78] was employed, using R package “glmnet” version 4.0-2 (Trevor Hastie). Parameters for glmnet included alpha = 1, family = binomial and the minimum lambda, which was determined by cross-validation, was used for prediction, whereas “class” was selected as a prediction measure. Metabolomics data were not scaled, whereas when including age and gender as variables data were autoscaled (unit variance). Optimal model parameters (number of latent variables in sPLSDA, number of included variables in sPLSDA and glmnet) were determined by cross-validation. Each model was assessed by double cross-validation (CV2) [79,80], by leaving out a number of samples determined by the k-fold method, with k = 8 and 7 for outer and inner loops, respectively. These values for k were switched to k = 6 and k = 5 (respectively) for approach 4 samples, where a higher class imbalance was present. In scenario CSVPHT, k = 7 and k = 6 (respectively), where a higher class imbalance was also present. A single repeat of the process was chosen for the inner loop and 50 repeats for the outer loop. Finally, the left-out sample labels were predicted based on the model and using the Mahalanobis distance. Each model’s signature was determined based on the total set of samples, after single CV and the same parameters as with CV2 (e.g., 50 repeats). The maximum number of latent variables for (s)PLSDA models was 10, and for each inner cross-validation loop, the minimum number of components with the maximum accuracy was chosen, based on predictions using the Mahalanobis distance. All parameters and scripts for multivariate data analysis can be found on the first author’s GitHub page (https://github.com/NickBliz/ENSAT-HT-PLASMA-NMR) and data can be provided upon request. Univariate statistical analysis methods were used to assess confounder effects on multivariate models and were performed using the “stats” R package [59], to check for data normality and to discover significant differences between variables. Tests included the ANOVA/Krusall–Wallis test, as well as the t/Wilcoxon test [81] to compare averages, at a significance level of 5%. For model accuracies, sensitivities and specificities reported in tables, one-sample *t*-tests were performed in R and the means and 95% confidence intervals were obtained. The Pearson/Kendall/Spearman correlation [82] estimates were used for investigating univariate correlations between various factors (technical, clinical and biological) and each model’s number of misclassifications. A *p*-value of less than 0.05 was accepted as statistically significant.

### 4.4. Confounders

Several approaches were used to investigate trends found by PCA: (A) Using ASCA [24], we estimated the confounder effects and used the Simultaneous Component Analysis residuals for sPLSDA. (B) We analyzed the data with PLSDA, based on groups defined by a specific confounder, to establish an inclusive list of all potentially relevant metabolites (i.e., without searching for most predictive metabolites, but rather all related metabolites, in our attempts to prove the study “innocent” of confounding effects). These metabolites were thereafter excluded from the dataset, in the initial phase of data processing (i.e., during the peak exclusion step), and the new dataset was used for classifying samples based on their disease groups. (C) Samples most related to the strongest confounder were removed, and classification models were calculated based on the remaining samples.

## 5. Conclusions

Our data reveal obvious between-center differences. Probably the most important factor is the retrospective multicenter character of the study. Specifically, for untargeted metabolomics studies, this stresses the importance of the preanalytical conditions (vena puncture- and sampling specifics, storage conditions and storage time, medication and dietary influences, transport and handling conditions, etc.). In the present study, the most likely explanation for the large variation in samples per center is a combination of pre-centrifugation and pre-storage delay, as well as variations in storage time, which form a triangle of effects with the disease group variation. These factors are equally important in prospective multicenter studies. In such studies, investigators can catch such variables in protocols and can use control samples as quality control means. In retrospective studies, multicenter quality controls may be more challenging to obtain, therefore, the study design should include a harmonized protocol, which would allow for these factors to be considered. This protocol should be followed by all participating centers [83] ensuring that preanalytical confounders are kept to a minimum. Samples should only be selected for inclusion if they fulfill certain preanalytical criteria. In fact, it would be beneficial for any multicenter or retrospective study to carry out a preliminary experiment on samples collected to assess the quality of the selected or prospective cohort. In addition, quality markers could help detect samples of low quality to be discarded [40]. Finally, normalization methods could be used for adjusting any residual preanalytical effects, provided that samples are selected in such a way that ensures their applicability [24,25,26,52]. Implementation of these procedures may prove vital in the discovery of robust disease biomarkers. Our study did not result in robust EHT biomarkers, due to the lack of adequate solutions and international consensus for containing the bias caused by preanalytical factors. This need should be covered by decisions on study design requirements for future multicenter metabolomics studies, with respect to future as well as published research findings on the effects of preanalytical conditions.

## Figures and Tables

**Figure 1 metabolites-12-00679-f001:**
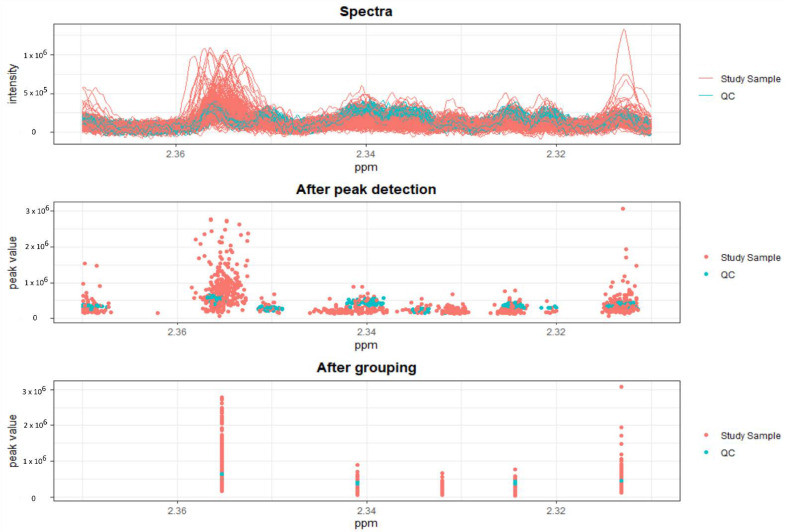
Results obtained after peak picking, as well after grouping and filling of the NMR spectra. Only the first half of samples analyzed are depicted.

**Figure 2 metabolites-12-00679-f002:**
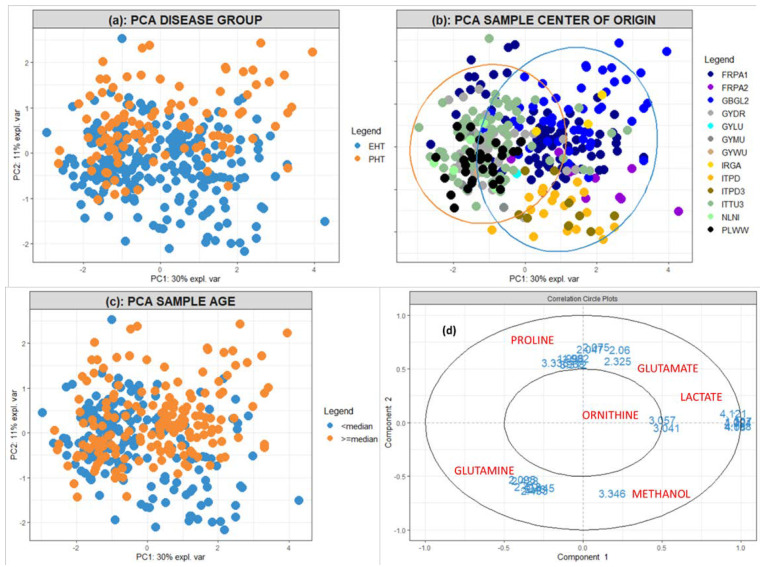
PCA plots of the first two principal components, calculated from all samples and all 86 peaks. In score plot (**a**), samples were colored according to disease group (CS, PA, PHT or PPGL), whereas in score plot (**b**), samples were colored according to the centers in which they were collected and score plot (**c**) depicts samples colored according to sample age, with the median value as the cutoff. In all scores plots, the percentage of explained variance per component is depicted in the plot axes. Though PC2 scores are slightly higher for PHT samples compared to EHT, samples were strikingly different from center to center. The 95% ellipses in plot (**b**) were calculated based on a score plot colored according to the two main clusters according to PC1, i.e., cluster 1 (orange) and cluster 2 (blue), and were included here to highlight what seems to be the most important source of variation in these data. Plot (**d**) is the loadings plot of the same PCA, with NMR peaks in blue and the corresponding metabolite names in red. Only peaks with a correlation cutoff above 0.5 are shown, as these arise from metabolites that most affect sample distribution in the scores plots (**a**–**c**).

**Figure 3 metabolites-12-00679-f003:**
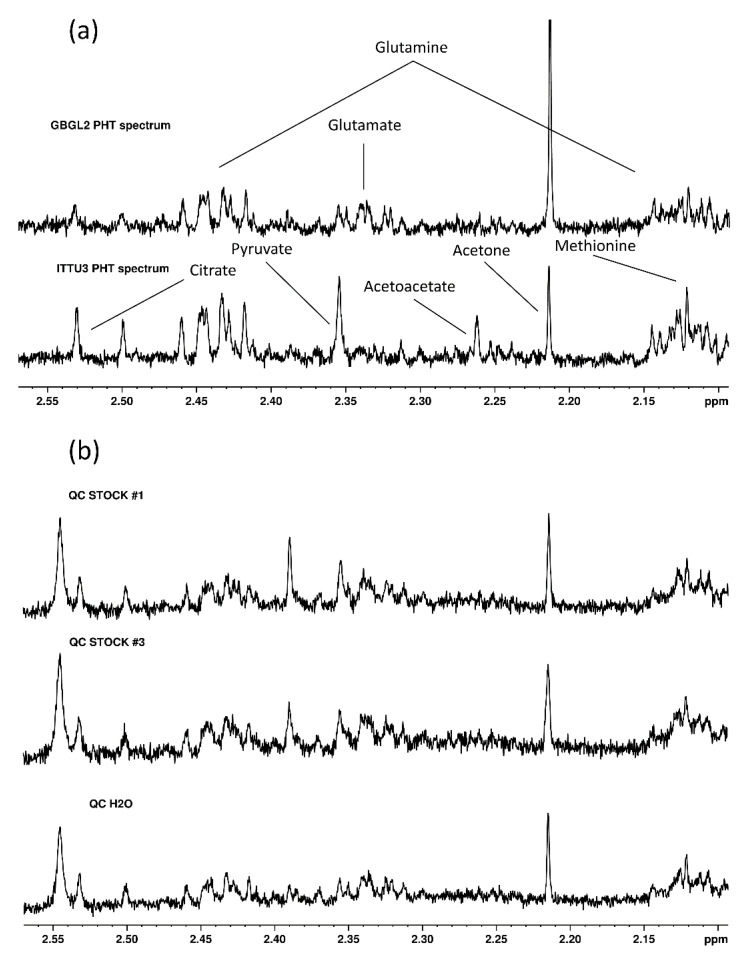
The identification of metabolites from NMR signals. After significant differences were detected and observed directly in spectra (**a**), spiking experiments were carried (**b**) out to validate assignments performed by means of 2D NMR experiments, namely J-resolved (**c**) and correlation spectroscopy (**d**).

**Table 1 metabolites-12-00679-t001:** Patient and sample characteristics. Abbreviations are explained in the main text.

	PHT (*n* = 106)	EHT (*n* = 231)	PA (*n* = 104)	PPGL (*n* = 94)	CS (*n* = 33)
PATIENT CHARACTERISTICS					
PATIENT AGE	55 [18,19,20,21,22,23,24,25,26,27,28,29,30,31,32,33,34,35,36,37,38,39,40,41,42,43,44,45,46,47,48,49,50,51,52,53,54,55,56,57,58,59,60,61,62,63,64,65,66,67,68,69,70,71,72,73,74,75,76,77,78,79] *	49 [17,18,19,20,21,22,23,24,25,26,27,28,29,30,31,32,33,34,35,36,37,38,39,40,41,42,43,44,45,46,47,48,49,50,51,52,53,54,55,56,57,58,59,60,61,62,63,64,65,66,67,68,69,70,71,72,73,74,75,76,77]	48 [26,27,28,29,30,31,32,33,34,35,36,37,38,39,40,41,42,43,44,45,46,47,48,49,50,51,52,53,54,55,56,57,58,59,60,61,62,63,64,65,66,67,68,69,70,71,72,73,74]	50 [19,20,21,22,23,24,25,26,27,28,29,30,31,32,33,34,35,36,37,38,39,40,41,42,43,44,45,46,47,48,49,50,51,52,53,54,55,56,57,58,59,60,61,62,63,64,65,66,67,68,69,70,71,72,73,74,75,76,77]	47 [17,18,19,20,21,22,23,24,25,26,27,28,29,30,31,32,33,34,35,36,37,38,39,40,41,42,43,44,45,46,47,48,49,50,51,52,53,54,55,56,57,58,59,60,61,62,63,64,65,66,67,68,69,70,71,72,73,74,75,76]
*p*-value **		**0.003**	**0.002**	0.1	**0.03**
PATIENT SEX (F/M)	61/45	135/96	47/57	58/36	30/3
*p*-value ***		0.9	0.1	0.6	**0.0003**
PREANALYTICAL SAMPLE CHARACTERISTICS					
SAMPLE AGE (days)	2393 [127–6418] *	1125 [11–3442] *	535 [52–2280] *	1548 [11–3442] *	162 [19–1186] *
*p*-value **		**6 × 10^−11^**	**4 × 10^−10^**	**0.002**	**3 × 10^−12^**
SAMPLE CENTER OF ORIGIN					
FRPA1	17 (16%)	66 (29%)	40 (38%)	26 (28%)	0
FRPA2	0	11 (4.8%)	0	0	11 (33%)
GBGL2	49 (46%)	0	0	0	0
GYDR	20 (19%)	28 (12%)	8 (7.7%)	19 (20%)	1 (3.0%)
GYLU	0	1 (0.4%)	0	1 (1.1%)	0
GYMU	0	4 (1.7%)	0	4 (4.3%)	0
GYWU	0	1 (0.4%)	0	1 (1.1%)	0
IRGA	0	3 (1.3%)	0	0	3 (9.1%)
ITPD	0	20 (8.7%)	2 (1.9%)	4 (4.3%)	14 (42%)
ITPD3	0	9 (3.9%)	8 (7.7%)	0	1 (3.0%)
ITTU3	20 (16%)	51 (22%)	46 (44%)	2 (2.1%)	3 (9.1%)
NLNI	0	6 (2.6%)	0	6 (6.4%)	0
PLWW	0	31 (13%)	0	31 (33%)	0
*p*-value ***		**5 × 10^−4^**	**5 × 10^−4^**	**5 × 10^−4^**	**6 × 10^−26^**
ANALYTICAL SAMPLE CHARACTERISTICS					
BATCH	22 [1,2,3,4,5,6,7,8,9,10,11,12,13,14,15,16,17,18,19,20,21,22,23,24,25,26,27,28,29,30,31,32,33,34,35,36,37,38,39,40,41,42] *	25 [1,2,3,4,5,6,7,8,9,10,11,12,13,14,15,16,17,18,19,20,21,22,23,24,25,26,27,28,29,30,31,32,33,34,35,36,37,38,39,40,41,42,43,44,45,46,47,48] *	22 [1,2,3,4,5,6,7,8,9,10,11,12,13,14,15,16,17,18,19,20,21,22,23,24,25,26,27,28,29,30,31,32,33,34,35,36,37,38,39,40,41,42,43,44]	31 [1,2,3,4,5,6,7,8,9,10,11,12,13,14,15,16,17,18,19,20,21,22,23,24,25,26,27,28,29,30,31,32,33,34,35,36,37,38,39,40,41,42,43,44,45,46,47,48] *	24 [1,2,3,4,5,6,7,8,9,10,11,12,13,14,15,16,17,18,19,20,21,22,23,24,25,26,27,28,29,30,31,32,33,34,35,36,37,38,39,40,41,42,43,44,45,46] *
*p*-value **		0.1	0.8	**0.006**	0.9
RUN ORDER	7 [2,3,4,5,6,7,8,9,10,11,12,13,14,15,16] *	7 [2,3,4,5,6,7,8,9,10,11,12,13,14,15,16,17] *	7 [2,3,4,5,6,7,8,9,10,11,12,13,14] *	9 [2,3,4,5,6,7,8,9,10,11,12,13,14,15,16,17] *	7 [2,3,4,5,6,7,8,9,10,11,12,13,14,15] *
*p*-value **		0.6	0.7	0.5	0.9

* Each continuous variable, such as patient age, is presented as a mean or median (depending on normality, indicated with the asterisk) and a range. ** For continuous variables, a *p*-value was obtained from a t/Wilcoxon test (depending on normality), comparing each disease group (Endocrine Hypertension (EHT), Primary Aldosteronism (PA), Pheochromocytoma/Paraganglioma (PPGL) or Cushing’s Syndrome (CS)) to the control group (Primary Hypertension, PHT). *** For categorical variables, such as patient sex, a *p*-value was obtained from a Fisher test, comparing each disease group (EHT, PA, PPGL or CS) to the control group (PHT).

**Table 2 metabolites-12-00679-t002:** Summary of accuracies from sPLSDA models via each approach for each scenario.

Scenario	Metric	Initial Approach	Approach A	Approach B	Approach C
EHT-PHT	Balanced Accuracy	79 (78–79)	79 (79–79)	67 (66–67)	58 (57–59)
	Sensitivity **	87 (87–87)	84 (83–84)	69 (69–70)	62 (61–63)
	Specificity ***	70 (70–71)	74 (73–74)	64 (63–65)	53 (51–55)
PA-PHT	Balanced Accuracy	83 (83–84)	83 (83–83)	69 (69–70)	69 (68–70)
	Sensitivity **	90 (89–90)	89 (89–90)	70 (69–71)	77 (76–79)
	Specificity ***	77 (77–78)	77 (77–77)	69 (68–70)	61 (60–62)
PPGL-PHT	Balanced Accuracy	79 (78–79)	81 (80–81)	68 (68–69)	68 (67–69)
	Sensitivity **	88 (87–88)	86 (85–87)	69 (68–70)	69 (68–70)
	Specificity ***	70 (69–70)	75 (75–76)	67 (66–68)	66 (65–68)
CS-PHT	Balanced Accuracy	85 (84–85)	-	82 (81–82)	-
	Sensitivity **	71 (71–72)	-	79 (78–80)	-
	Specificity ***	98 (98–99)	-	84 (84–85)	-
ALL-ALL	Balanced Accuracy	65 (64–65)	-	53 (52–53)	57 (57–58)
	CS TP Rate	73 (72–74)	-	72 (71–73)	-
	PA TP * Rate	65 (64–65)	-	53 (52–54)	69 (67–70)
	PHT TP * Rate	72 (72–72)	-	45 (45–46)	35 (33–36)
	PPGL TP * Rate	50 (49–51)	-	42 (41–43)	69 (68–70)

* TP stands for True Positive. ** Sensitivity is the TP rate of the disease group (EHT, PA, PPGL or CS). *** Specificity is the TP rate of the control group (PHT). All metrics are given as means, with the 95% confidence interval (in brackets).

**Table 3 metabolites-12-00679-t003:** Regression coefficients for peaks representative of each metabolite, obtained via sPLSDA on the EHT vs. PHT scenario, from each approach. Most peaks that were selected as predictors via the Initial Approach were not selected via Approach C. Lactate, which was selected by both, has a negative coefficient in the Approach C model, contrasting the Initial Approach. Metabolites highlighted in **bold** were found to have a strong relationship with a confounder (Table 4).

Metabolite	NMR Signal (ppm)	Initial Approach	Approach A	Approach B	Approach C
Alanine	1.457	−0.19975	−0.07187	0	0
**Creatine**	3.917	0	0		0.019157
Creatinine	4.041	0	0	0.177419	0
**Dimethyl sulfone**	3.137	0	0		0.109629
Dimethylamine	2.695	0	0	−0.03439	0
Dimethylglycine	2.91	0	0	0.04487	0.027703
Formate	8.441	−0.01988	−0.01755	0.023133	0
**Glutamine**	2.433	0.148614	0.135957		0
**Glutamate**	2.325	−0.12554	−0.14981		0
**Glucose**	5.22	0.039396	0.0097		0.147391
**Glycine**	3.548	−0.0108	0		0
**Glycerol**	3.555	0	0		0.091654
**Lactate**	4.108	0.025885	0		−0.08734
Lysine	2.997	0	0	0.03347	0
**Methionine**	2.122	0.052659	0.02404		0
**Methanol**	3.346	0.062726	0.050343		0.04658
Proline	1.996	−0.02291	−0.00628	−0.13954	−0.01636
**Pyruvate**	2.356	0.312859	0.32791		0.197295
Threonine	4.24	0	0	0.040696	0
Tyrosine	7.168	0	0	−0.0194	0
Valine	0.981	0	0	0	−0.00058
Unknown Metabolites	3.162	0.009448	0.017788	0.236056	0
	3.262	0	0	−0.15528	−0.05692
	3.284	−0.03909	−0.02878		0
	3.612	0	0	−0.11482	0
	3.67	0	0	−0.12957	0

**Table 4 metabolites-12-00679-t004:** Metabolites selected for exclusion due to a strong relationship with a confounder.

Metabolite	NMR Peaks (ppm)	Dataset	Reason *	FRPA1 PHT/Cluster 2/High Sample Age
Acetylcarnitine	3.177	PA-PHT, PPGL-PHT	PLSDA CLUSTER, SAMPLE AGE	↓
Creatine	3.021, 3.917	PA-PHT, PPGL-PHT	PLSDA CLUSTER, SAMPLE AGE	↑
Dimethyl sulfone	3.137	PA-PHT, PPGL-PHT	PLSDA SAMPLE AGE	↑
Glucose	5.220, 5.227	PA-PHT, PPGL-PHT	PLSDA CLUSTER, SAMPLE AGE	↓
Glutamate	2.047, 2.060, 2.075, 2.095, 2.103, 2.108, 2.113, 2.122, 2.132, 2.140, 2.145, 2.325, 2.332, 2.341, 2.356	PA-PHT, PPGL-PHT	FRPA1 PHT, PLSDA CLUSTER, SAMPLE AGE	↑
Glutamine	2.095, 2.103, 2.108, 2.113, 2.122, 2.132, 2.140, 2.145, 2.418, 2.428, 2.433, 2.444, 2.449, 2.460	PA-PHT, PPGL-PHT	FRPA1 PHT, PLSDA CLUSTER, SAMPLE AGE	↓
Glycerol	3.555, 3.567	PA-PHT	PLSDA CLUSTER, SAMPLE AGE	↑
Glycine	3.548	PA-PHT, PPGL-PHT	PLSDA SAMPLE AGE	↓
Lactate	1.321, 1.307, 4.080, 4.094, 4.108, 4.121	PA-PHT, PPGL-PHT	PLSDA CLUSTER, SAMPLE AGE	↑
Methanol	3.346	PA-PHT, PPGL-PHT	PLSDA CLUSTER, SAMPLE AGE	↓
Methionine	2.122	PA-PHT, PPGL-PHT	FRPA1 PHT, PLSDA SAMPLE AGE	↓
Ornithine	3.041, 3.057	PA-PHT, PPGL-PHT	PLSDA CLUSTER, SAMPLE AGE	↑
Pyruvate	2.356	PA-PHT, PPGL-PHT	PLSDA CLUSTER, SAMPLE AGE	↓
Unknown metabolite	3.284	PA-PHT, PPGL-PHT	PLSDA CLUSTER, SAMPLE AGE	↑

* Peaks were excluded either because they were found to be important in discriminating samples in a Partial Least Squares Discriminant Analysis (PLSDA) of center cluster, i.e., the separation of centers according to the first dimension in the PCA score plot of Figure 2b, in a PLSDA of sample age (with the median sample age as a cutoff), or because they were found to be higher or lower in the FRPA1 PHT group of samples.

**Table 5 metabolites-12-00679-t005:** Patient and sample characteristics, after whole center exclusions (Approach C). Abbreviations are explained in the main text.

	PHT (*n*= 40)	EHT (*n* = 118)	PA (*n* = 54)	PPGL (*n* = 64)
PATIENT CHARACTERISTICS				
PATIENT AGE	44 [19,20,21,22,23,24,25,26,27,28,29,30,31,32,33,34,35,36,37,38,39,40,41,42,43,44,45,46,47,48,49,50,51,52,53,54,55,56,57,58,59,60,61,62,63,64,65,66,67,68,69,70]	49 [19,20,21,22,23,24,25,26,27,28,29,30,31,32,33,34,35,36,37,38,39,40,41,42,43,44,45,46,47,48,49,50,51,52,53,54,55,56,57,58,59,60,61,62,63,64,65,66,67,68,69,70,71,72,73,74]	48 [32,33,34,35,36,37,38,39,40,41,42,43,44,45,46,47,48,49,50,51,52,53,54,55,56,57,58,59,60,61,62,63,64,65,66,67,68,69,70,71,72,73,74]	50 [19,20,21,22,23,24,25,26,27,28,29,30,31,32,33,34,35,36,37,38,39,40,41,42,43,44,45,46,47,48,49,50,51,52,53,54,55,56,57,58,59,60,61,62,63,64,65,66,67,68,69,70,71,72,73,74]
*p*-value **		**0.03**	0.05	**0.04**/0.6
PATIENT SEX (F/M)	15/25	65/53	24/30	41/23
*p*-value ***		0.07	0.5	**0.009/0.04**
PREANALYTICAL SAMPLE CHARACTERISTICS				
SAMPLE AGE	366 [127–1307] *	748 [83–2841] *	380 [83–1598] *	1419 [121–2841]
*p*-value **		**1 × 10^−5^**	1	**4 × 10^−13^/1 × 10^−15^**
SAMPLE CENTER OF ORIGIN				
GYDR	20 (50%)	27 (23%)	8 (15%)	19 (30%)
GYLU	0	1 (0.8%)	0	1 (1.6%)
GYMU	0	4 (3.4%)	0	4 (6.3%)
GYWU	0	1 (0.8%)	0	1 (1.6%)
ITTU3	20 (50%)	48 (41%)	46 (85%)	2 (3.1%)
NLNI	0	6 (5.1%)	0	6 (9.4%)
PLWW	0	31 (26%)	0	31 (48%)
*p*-value ***		**6 × 10^−5^**	**5 × 10^−4^**	**2 × 10^−13^/4 × 10^−23^**
ANALYTICAL SAMPLE CHARACTERISTICS				
BATCH	19 [1,2,3,4,5,6,7,8,9,10,11,12,13,14,15,16,17,18,19,20,21,22,23,24,25,26,27,28,29,30,31,32,33,34,35,36,37,38,39,40,41,42]	27 [1,2,3,4,5,6,7,8,9,10,11,12,13,14,15,16,17,18,19,20,21,22,23,24,25,26,27,28,29,30,31,32,33,34,35,36,37,38,39,40,41,42,43,44,45,46,47,48] *	22 [1,2,3,4,5,6,7,8,9,10,11,12,13,14,15,16,17,18,19,20,21,22,23,24,25,26,27,28,29,30,31,32,33,34,35,36,37,38,39,40,41,42]	37 [1,2,3,4,5,6,7,8,9,10,11,12,13,14,15,16,17,18,19,20,21,22,23,24,25,26,27,28,29,30,31,32,33,34,35,36,37,38,39,40,41,42,43,44,45,46,47,48] *
*p*-value **		**0.001**	0.1	**7 × 10^−5^/0.0001**
RUN ORDER	7 [2,3,4,5,6,7,8,9,10,11,12,13,14,15,16] *	9 [2,3,4,5,6,7,8,9,10,11,12,13,14] *	8 [2,3,4,5,6,7,8,9,10,11,12,13,14] *	9 [2,3,4,5,6,7,8,9,10,11,12,13,14] *
*p*-value **		0.3	0.4	0.4/1

* Each continuous variable, such as patient age, is presented as a mean or median (depending on normality, indicated with the asterisk) and a range. ** For continuous variables, a *p*-value was obtained from a t/Wilcoxon test (depending on normality), comparing each disease group (EHT, PA, or PPGL) to the control group (PHT). The PPGL column has an additional *p*-value obtained from the comparison of PA to PPGL. *** For categorical variables, such as patient sex, a *p*-value was obtained from a Fisher test, comparing each disease group (EHT, PA, or PPGL) to the control group (PHT). The PPGL column has an additional *p*-value obtained from the comparison of PA to PPGL.

## Data Availability

Due to sensitive patient information, data were made available on zenodo under restricted access (doi:10.5281/zenodo.6614995), in the folder “Preanalytical Pitfalls in Untargeted Plasma Nuclear Magnetic Resonance Metabolomics of Endocrine Hypertension”.

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
