# Peer review of "Preanalytical Pitfalls in Untargeted Plasma Nuclear Magnetic Resonance Metabolomics of Endocrine Hypertension"

_metabolites, 2022, doi:10.3390/metabo12080679_

Round 1
Reviewer 1 Report
In this study titled 'Preanalytical Pitfalls in Untargeted Plasma Nuclear Magnetic 2 Resonance Metabolomics of Endocrine Hypertension', the authors Bliziotis et al. analyzed plasma samples for metabolic profiling from various centers with proton NMR. Using three different approaches, the authors aim to establish unbiased endocrine hypertension biomarkers. In general, I found this work to be well conducted, and have significant impact on ETH metabolomics. Some major revisions are needed. See detailed comments below.
Major points:
1. The raw NMR data are not available, thus making it impossible to assess the data quality. The authors should consider making all the raw NMR spectra (including 1D and 2D data) available by including them in the supplementary materials or deposit them into a database if the files are too big.
2. The authors should demonstrate in at least one figure that how the NMR spectra were deconvoluted and assignment of NMR peaks were performed.
3. The authors need to provide more information regarding the PCA analyses. What are the principle components used in these PCA plots? All the metabolites or a subset of markers used? What do the x- and y- axes (30% expl. var and 11% expl. var) stand for?
4. What could explain the large variation in samples from different centers? Different sample collection/storage protocols?
5. What is the median value of sample age?
6. It would be of interest to see the separation between different disease groups with a non-diseased control group.
7. One major limitation of this study is the lack of information of patient samples (e.g. age, gender, and disease severity), the variation observed in this study may reflect some of these factors. The authors should give some thoughts on these factors.
8. Given the results presented in this paper, the authors might want to give some discussion on how to avoid the confounders and pitfalls in ETH metabolic profiling (e.g. standard protocols in sample handling/storage, certain age limit for sample, etc).
Minor:
1. Full names of abbreviations should be introduced the first time they appear in the text. e.g. PCA, QC, etc.
2. line 139, 'a low analytical compared' ?
3. The panels in Fig.1 should be organized in a better way.
Reviewer 2 Report
In this study, the authors compared the H1 NMR spectra of 106 PHT patients to 231 EHT patients using sPLSDA models. They investigated the confounding effects, and corrected them by ASCA, Metabolite exclusions, and Whole center exclusions. They found that the between-center differences is the the most important confounding effect. They proposed to use the multicenter quality controls to ensure the confounders as minimum as possible. The following issues need to be addressed:
(1). The numbers of EHT and PHT were 231 and 106, respectively. How to deal with the category imbalance problem?
(2). In the PCA plot, the overlap between the PHT and EHT samples was so severe that it could not be seen to successfully separate the two types of samples, PHT and EHT.
(3). The 1st and 2nd principal components explained only 41% of the variance in the data.
(4). The explained variance of PC1 in Figure 1B is covered by Figure 1C.
(5). The four subgraphs in Figure 1 are too large and of different sizes.
(6). The authors found some important variables based on the Loading diagram of PCA and listed the compound names according to the chemical shifts. These compounds are relatively common (Except that methanol is not commonly found in plasma), do they have biological significance with hypertension?
(7). It is necessary to verify that the sPLSDA method is not overfitted using the permutation test.
Reviewer 3 Report
The manuscript entitled “Preanalytical Pitfalls in Untargeted Plasma Nuclear Magnetic Resonance Metabolomics of Endocrine Hypertension” by Nikolaos G. Bliziotis et al. employed untargeted NMR metabolomics to find the potential biomarkers for differentiating endocrine hypertension (EHT) forms. The authors have analyzed the 1H-NMR spectra of patients from different countries. After various statistical analysis, the authors concluded that, due to the confounding effect, the untargeted NMR metabolomics cannot establish a reliable EHT-PHT signature. The authors have given in-depth discussion about the factors that significantly contribute to the confounding effect, such as preanalytical conditions, age, medication status. The authors have also discussed the limitations of this study and provided suggestions for further research. Overall, the manuscript provided rich information about the metabolomic studies of EHT and PHT. I suggest to accept to publish on Metabolites. However, the writing of the manuscript needs to be improved to increase the readability.
Round 2
Reviewer 1 Report
All my comments are adequately addressed.
Reviewer 2 Report
The authors have revised the manuscript according to previous comments, and I have no further comments.